# Real-World Effectiveness of Mix-and-Match Vaccine Regimens against SARS-CoV-2 Delta Variant in Thailand: A Nationwide Test-Negative Matched Case-Control Study

**DOI:** 10.3390/vaccines10071080

**Published:** 2022-07-05

**Authors:** Rapeepong Suphanchaimat, Natthaprang Nittayasoot, Chuleeporn Jiraphongsa, Panithee Thammawijaya, Punsapach Bumrungwong, Atthavit Tulyathan, Nontawit Cheewaruangroj, Chakkarat Pittayawonganon, Piyanit Tharmaphornpilas

**Affiliations:** 1Division of Epidemiology, Department of Disease Control, Ministry of Public Health, Nonthaburi 11000, Thailand; n.natthaprang@gmail.com (N.N.); jiraphongsa@gmail.com (C.J.); viewfetp@gmail.com (P.T.); c.pittayawonganon@gmail.com (C.P.); piyanit@health.moph.go.th (P.T.); 2International Health Policy Program, Ministry of Public Health, Nonthaburi 11000, Thailand; 3Government Big Data Institute, Bangkok 10900, Thailand; punsapach.bu@depa.or.th (P.B.); atthavit.tu@depa.or.th (A.T.); nontawit.ch@depa.or.th (N.C.)

**Keywords:** COVID-19, SARS-CoV-2, vaccine effectiveness, Delta variant, Thailand

## Abstract

The objective of this study is to explore the real-world effectiveness of various vaccine regimens to tackle the epidemic of severe acute respiratory syndrome coronavirus 2 (SARS-CoV-2) Delta variant in Thailand during September–December 2021. We applied a test-negative case control study, using nationwide records of people tested for SARS-CoV-2. Each case was matched with two controls with respect to age, detection date, and specimen collection site. A conditional logistic regression was performed. Results were presented in the form vaccine effectiveness (VE) and 95% confidence interval. A total of 1,460,458 observations were analyzed. Overall, the two-dose heterologous prime-boost, ChAdOx1 + BNT162b2 and CoronaVac + BNT162b2, manifested the largest protection level (79.9% (74.0–84.5%) and 74.7% (62.8–82.8%)) and remained stable over the whole study course. The three-dose schedules (CoronaVac + CoronaVac + ChAdOx1, and CoronaVac + CoronaVac + BNT162b2) expressed very high degree of VE estimate (above 80.0% at any time interval). Concerning severe infection, almost all regimens displayed very high VE estimate. For the two-dose schedules, heterologous prime-boost regimens seemed to have slightly better protection for severe infection relative to homologous regimens. Campaigns to expedite the rollout of third-dose booster shot should be carried out. Heterologous prime-boost regimens should be considered as an option to enhance protection for the entire population.

## 1. Introduction

Coronavirus disease 2019 (COVID-19) is now recognized as one of the most serious health threats in human history. The causative pathogen of COVID-19 is severe acute respiratory syndrome coronavirus 2 (SARS-CoV-2) [1,2]. In the latter half of 2021, the world was severely hit by the SARS-CoV-2 Delta variant. In June 2021, the World Health Organization (WHO) reported that the Delta variant became the dominant strain globally [3,4]. At the time of writing, the global case toll has exceeded 416 million with approximately 5.8 million cumulative deaths [5].

Thailand is among numerous countries severely suffering from the Delta wave. The first COVID-19 wave in Thailand was caused by the original SARS-CoV-2 strain during March–May 2020, followed by the second Alpha wave originated from a cluster of cases in the inner city of a vicinity province of Bangkok [6,7]. The third wave in April 2021 was still attributed to the Alpha variant. Then, the country was hardest hit by the fourth wave, caused by the Delta variant, in the second half of 2021 [7]. During that time, the Thai government implemented a lockdown policy to reduce the case and death tolls. Apart from aggressive social measures, vaccine rollout was deemed as an ultimate weapon to halt the pandemic. The journey of COVID-19 national immunization plan in Thailand commenced in February 2021 when the government imported CoronaVac from China to alleviate the rise of cases in response to the second wave. The initial plan of the government was using domestically produced viral vector vaccine, ChAdOx1, as the dominant vaccine for the Thai population. However, the advent of the Delta wave prompted a huge demand for vaccines, far outstripping the pace of domestic production and there was significant global concern that viral vector or inactivated viral vaccines might be less effective in tackling the Delta variant, compared with mRNA vaccines [8,9]. Another compelling reason that required the government to adjust the national vaccine plan was evidence suggesting that the immunity level of the previously immunized population by CoronaVac in early 2021 rapidly declined in the first few months [10].

To this end, the government adjusted the national immunization plan by purchasing a huge bulk of various vaccine types, including mRNA vaccines (BNT162b2), in combination with an acceleration of domestic vaccine production. In addition, massive campaigns for booster (third) dose and proposals of mix-and-match vaccine schedules attracted remarkable academic and political attention [11]. By late 2021, the National Vaccine Committee (NVC) approved mix-and-match vaccine schedules, starting with CoronaVac as the first dose followed by ChAdOx1 as the second dose (one month apart), since domestic study suggested comparable immunogenicity with the standard two-dose regimen of ChAdOx1, which required a three-month interval between doses [12]. So far, the government has endorsed many more mix-and-match vaccine regimens in the current national COVID-19 immunization plan.

Though various vaccine regimens have been applied in the field, little is known about the real-world effectiveness at the nationwide scale. Moreover, epidemiological research on heterologous vaccine regimens is quite sparse. Therefore, the objective of this study is to determine the effectiveness of various vaccine regimens during the Delta wave in Thailand (September–December 2021) using real-world immunization data for the whole Thai population.

## 2. Materials and Methods

### 2.1. Study Design

We applied a test-negative matched case-control design.

### 2.2. Data Sources

The data retrieval process consisted of three steps. First, we explored the Co-Lab database, which is the national laboratory recording system of the Department of Medical Sciences (DMSc), Ministry of Public Health (MOPH). Both public and private health facilities reported health service records of people undertaking the polymerase chain reaction (PCR) test for SARS-CoV-2 to the Co-Lab. By late 2021, the national guideline for COVID-19 diagnosis allowed rapid antigen diagnostic test (Ag-RDT) to replace PCR where PCR capacity is limited. Therefore, about 10–20% of the cases stored in the Co-Lab system were identified by Ag-RDT instead of PCR. During that period, the official count of Ag-RDT positive cases was limited to professional-use only. Thus, a person with positive Ag-RDT self-test was not included in this study.

A case was defined as a Thai national with positive SARS-CoV-2 by PCR test between 1 September 2021 and 31 December 2021 while a control was defined as a Thai national showing negative PCR test for SARS-CoV-2 or professional use Ag-RDT in the same period. We selected all cases and identified two controls per case. The matching of triples (a case and two controls) was performed with respect to age (allowing a three-year margin), laboratory detection date (allowing a seven-day margin), and provincial residence of testing sites (exact match). Apart from the test result, the Co-Lab database also provided reasons for each test record (such as for contact tracing due to being high-risk contact of a COVID-19 case, for active case finding, and for other reasons). Then, to retrieve information about illness severity of each case, we joined the Co-Lab database with the other two databases, namely, Co-Ward database and the COVID-19 Death database. The Co-Ward database, managed by the Department of Medical Services (DMS), is the national surveillance system to monitor clinical severity and hospital bed capacity. The COVID-19 Death database, governed by the Department of Disease Control (DDC), is the national monitoring system for all COVID-19 related deaths. We obtained the immunization history (vaccination date, number of vaccines, and type of vaccines) of each individual from the MOPH Immunization Centre (MOPH-IC).

We combined cases and controls appearing in the aforementioned databases (by using an encrypted national identification number, an official identity for all Thai nationals, as a primary key to link same individuals across databases). Moreover, we excluded cases (and their matched controls) whose laboratory collection occurred within fourteen days after the last vaccination date to avoid ambiguity of the vaccine status since global evidence suggests that immune response needs about fourteen days after the last shot to have adequate protective effect against the virus [13]. Finally, about 1.5 million records were included in the analysis for VE estimate. A summary of the data retrieval process is given in Figure 1.

For more details, we began with selecting two controls per case since 1 July 2021. Then, we dropped records during July–August 2021 to avoid the influence of the Alpha variant–the dominant strain nationwide during the first half of 2021. By allowing a seven-day margin in the matching process, the drop of records during July–August 2021 made the ratio of a case per controls approximately equate 1:2 instead of keeping the exact ratio of 1:2.

### 2.3. Data Analysis

We started with an overview of the data by descriptive statistics. Then, we applied conditional logistic regression to estimate the odds of infection in the vaccinees (for all brands combined and for specific regimens) relative to the odds in the unvaccinated group. The findings appeared in the form of odds ratio (OR) and 95% confidence interval (CI). For communication convenience, we present the results in the form of vaccine effectiveness (VE) and 95% CI where VE equated one minus OR.

All VE estimates were determined in two strands: against any infection, and against severe infection. For the VE estimate against any infection, controls were defined as non-infectee samples. For the VE estimate against severe infection, a case was classified as samples undergoing severe infection or death while controls comprised a combination of non-infectees and non-severe infectees. Note that a severe case for this study was defined as a person experiencing hypoxemic pneumonia or an intubated patient or a death.

We also assessed if the main analysis was still robust if the observations were restricted to high-risk samples. The high-risk samples were defined as people undertaking SARS-CoV-2 test for contact tracing and active case finding.

We later examined if the VE waned over time by dividing the analysis into four periods according to time since the last vaccination date: 15–29 days, 30–59 days, 60–89 days, and 90 days onward. Subsequently, we examined the VE for each vaccine regimen over time with a special attention on the two-dose and three-dose schedules which were widely distributed at that time. For the two-dose vaccinees, we focused on the following regimens: BNT162b2 + BNT162b2, ChAdOx1 + ChAdOx1, CoronaVac + CoronaVac, ChAdOx1 + BNT162b2, CoronaVac + ChAdOx1, and CoronaVac + BNT162b2. For the three-dose schedule, we focused on CoronaVac + CoronaVac + ChAdOx1 and CoronaVac + CoronaVac + BNT162b2.

## 3. Results

We obtained a total of 1,698,588 records (558,865 cases and 1,139,723 controls). People aged between 18 and 59 years constituted the majority of the study participants. About 1.8% of the cases developed severe symptoms and half of the severe cases (50.2%) were the elders (>60 years). Forty-three percent of the participants were classified as high-risk samples. Almost a fifth of the participants undertook testing in Bangkok and five adjacent provinces (Nakhon Pathom, Pathum Thani, Nonthaburi, Samut Prakan, and Samut Sakhon), so-called Greater Bangkok, Table 1.

From about 1.7 million records, we dropped approximately 240,000 records where the laboratory collection date of cases (and the matched controls) occurred within fourteen days after the most recent vaccination date. As a result, about 1.4 million samples remained in the analysis. We later analyzed these samples by conditional logistic regression. For these samples, about a third of the participants were unvaccinated. Participants receiving two doses constituted the greatest share of overall vaccinees. Approximately two-thirds (66.1%) of the severe cases were unvaccinated. Only 8.8% of the participants had received the third shot. The percentage of three-dose vaccinees was more pronounced in the control group (12.0%), Table 2.

The volume of participants declined over time, from more than 380,000 controls and about 170,000 cases in September to approximately 137,000 controls and 76,000 cases in December. The percentage of two-dose vaccinees grew substantially from 20.5% to 61.9% throughout the study period. By December, the proportion of three-dose vaccinees was 13.1%, far larger from the proportion in September (6.2%). More details are presented in Appendix A.

Figure 2 demonstrates that the three-dose vaccination, regardless of the regimens, provided a high level of VE estimates for both any infection (90.3% (90.0–90.5%)) and severe infection protections (98.3% (97.6–98.8%)). The VE estimate against any infection for a single shot was 9.7% (8.6–10.8%). Two-dose vaccination exhibited a moderate degree of protection against any infection (45.9% (45.4–46.5%)) but the protection effect against severe infection was still high (85.4% (84.0–86.6%)). The restricted sample analysis on high-risk samples followed the same pattern as full sample analysis. In general, the restricted samples demonstrated slightly higher VE estimates than the full samples.

The analysis tallied by time since the last shot found that the VE estimate for three-dose vaccination for all regimens combined remained high (over 80%) with a negligible decline over time for both any and severe infections. The most distinctive protection benefit was found in severe infection protection among three-dose vaccinees (98–99% over time). In contrast, a remarkable drop of VE estimate for two-dose vaccination was observed. The most obvious waning of VE estimate was found in any infection analysis where the estimate dropped from 54.1% (53.2–55.1%) before day 30 to 40.3% (38.8–41.8%) after day 90. A decline of VE estimate against severe protection was also observable but this was less evident compared to the estimate against any infection, Figure 3.

The breakdown analysis found that all regimens exhibited varying degree of the VE estimate over time. Overall, heterologous prime-boost, ChAdOx1 + BNT162b2 and CoronaVac + BNT162b2, manifested the largest protection level (79.9% (74.0–84.5%) and 74.7% (62.8–82.8%)) within 30 days, and relative stable VE until day 90 and onward. BNT162b2 + BNT162b2 showed a protection level of 74.2% (71.8–76.3%) within 30 days but declined to 57.0% (43.6–67.2%) on day 90 and onward. ChAdOx1 + ChAdOx1 and CoronaVac + ChAdOx1 provided initial moderate protection level and declined relatively quick over time whereas CoronaVac + CoronaVac provided moderate protection after day 30 and onward. The three-dose schedules (CoronaVac + CoronaVac + ChAdOx1, and CoronaVac + CoronaVac + BNT162b2) expressed very high degree of VE estimate (above 80.0% at any time interval), Table 3. Note that the time lag between laboratory collection date and last vaccination date for the vaccinees whose laboratory collection date occurred at least 90 days after the last vaccination date did not vary much by vaccine regimens as detailed in Appendix A.

Concerning severe infection, in general, almost all regimens displayed very high VE estimate (highest at 99.1% and lowest at 80.3%). For the two-dose regimens, heterologous prime-boost seemed to have slightly better protection for severe infection compared with homologous regimens. The three-dose vaccinees benefited from the vaccine by about 97.2–99.1% effectiveness against severe infection. The largest protection effect was observed in CoronaVac + CoronaVac + ChAdOx1 between day 30 and day 59. The waning of VE estimate was minimal in all regimens, compared with the analysis on any infection. Of note is that, in certain time intervals, there were fewer than 10 severe cases amongst the vaccinees. As a result, accurate VE estimate could not be determined (hence we specified the VE estimate in that period as “not applicable”). For instance, there was only one severe case out of 435 vaccinees (0.2%) of CoronaVac + BNT162b2 in the earliest time interval (15–29 days), Table 4.

## 4. Discussion

This study is probably one of the very first studies on COVID-19 vaccine effectiveness using real-world service data in southeast Asia, and perhaps in Asia. Overall, the full (two-dose) vaccination (regardless of the vaccine brands) contributed to moderate degree of protection against SARS-CoV-2 Delta variant infection by approximately 50%, while the three-dose regimens provided about 90% effectiveness. The two-dose effectiveness from our findings was fairly inferior to the finding from recent meta-analysis by Zheng et al., which suggested an 89.1% estimate among fully vaccinated individuals [14]. Such a difference was likely due to dissimilarity in the analysis period between the two studies. Zheng et al. gathered literature published during August 2020–October 2021, the period before the Delta variant prevailed across the globe. In contrast, our study focused on the latter half of 2021, when the Delta variant constituted the dominant share of all SARS-CoV-2 variants globally [15].

Although the effectiveness against any infection of the two-dose regimen seemed to be mediocre, the value of severe infection protection was still obvious (85.4% (84.0–86.7%)). This justifies the merit of massive and speedy vaccine rollout in the population. Evidence from many nations also confirmed this. For example, Haas et al. reported that the rapid mass roll-out of the Pfizer-BioNTech vaccine in early 2021 helped reduce thousands of deaths and hospitalizations; and, combined with strict non-pharmaceutical interventions, the massive vaccine rollout contributed to the rebound of the Israeli economy by 5.5% in 2021 [16]. Suthar et al. found that, in the United States (US), during the first half of 2021, when the Alpha variant of SARS-CoV-2 took a lion share, the COVID-19 mortality rate diminished by 81% in counties with high vaccine coverage, compared with counties that had very low coverage. The impact on mortality followed the same pattern during the second half of 2021 when the Delta variant became a dominant strain, despite smaller effects on the case incidence [17]. By late 2021, the Thai government set mass vaccine rollout as a national agenda to address the pandemic. Yet, several challenges remain as massive immunization is not just a matter of individual propensity to accept the vaccine, but is also involved with many system angles, such as affordability, allocation, deployment, and production capacity [18].

This study affirmed the benefit of a COVID-19 vaccine booster shot though no perfect protection against breakthrough infection. The VE estimate against any infection of the three-dose regimen varied about 90–95% with little waning over time. In contrast, for two-dose vaccination, the effectiveness against any infection markedly fell as time passed by. The decline of immunity was also observed in many studies abroad. Goldberg et al. indicated that, in Israeli residents, the immunity against the Delta variant of SARS-CoV-2 waned in all age groups a few months after receipt of the second dose of BNT162b2 [19]. For the three-shot individuals, the effectiveness saw a minimal decline and the overall effectiveness throughout the six-month period was about 92% against any infection and 99% against severe infection and death. This discovery coincided with many studies from Europe and the US, which corroborated the value of the booster shot [20,21,22]. Thompson et al. indicated a 94% effectiveness against hospitalization, fourteen days after the third shot of BNT162b2 [21]. A study in Israel by Barda et al. pointed to a 93% effectiveness against hospitalization for individuals receiving the third dose of BNT162b2 [20]. A study in the United Kingdom (UK) by Andrews et al. suggested that the relative effectiveness against symptomatic infection about a month after taking BNT162b2 or mRNA-1273 (Moderna) booster shot with the use of ChAdOx1-S and BNT162b2 as a primary course varied between 85% and 95% [22]. Domestic evidence by Kanokudom et al. and Yorsaeng et al. revealed higher neutralizing activity against all variants of concern of SARS-CoV-2 amongst the recipients of a third dose of ChAdOx1 (after two-dose CoronaVac) than those completing two-dose CoronaVac or ChAdOx1 alone [23,24]. It is worth noting that though the merit of the booster shot is apparent, further research is still needed to explore the proper timing of receiving the booster shot. It is possible that people are advised to receive annual COVID-19 vaccination as long as SARS-CoV-2 continues to circulate within the global population.

A notable discovery in this study is that heterologous prime-boost regimens (especially, ChAdOx1 or CoronaVac followed by BNT162b2) provided favorable protection benefit which was relatively stable over time compared with homologous regimens, including BNT162b2 + BNT162b2. There is increasing international interest in heterologous prime-boost COVID-19 regimens to mitigate supply shock or shortage of vaccines that might otherwise reduce the speed of vaccine rollout [25,26]. Recent studies, although few in number at the time of writing, pointed to the same direction that mix-and-match regimens, if exercised appropriately, can serve as another powerful tool to combat the pandemic. A prospective cohort immunogenicity study in Thailand found that receptor-binding domain (RBD)-specific antibody responses against wild-type and variants of concern of SARS-CoV-2 were higher in the heterologous CoronaVac + ChAdOx1 and homologous ChAdOx1 + ChAdOx1 regimens in comparison with CoronaVac + CoronaVac regimen [27]. Another study from China revealed that heterologous prime-boost strategy significantly enhanced neutralizing antibody titers and improved T-helper 1 (TH1) responses [28]. A large nationwide cohort study in Sweden estimated that using ChAdOx1 as the first dose, followed by either the BNT162b2 or mRNA-1273 as the second dose, resulted in 67% and 79% effectiveness against symptomatic COVID-19 infection, respectively [29]. Recent evidence also suggested that the prime–boost schedules showed mild adverse events and favorable safety, comparable to the homologous counterparts [27,30,31].

The mix-and-match of vaccines from different platforms has long been practiced before the advent of SARS-CoV-2. A number of possible mechanisms for the higher immunity caused by heterologous vaccine schedules have been proposed. The underlying mechanism for higher immunity of heterologous prime-boost schedules has not been clearly described. However, it is possible that by using dissimilar vaccine formulations, different arms of the immune system are evoked. As a result, a combination of cellular and humoral immunity engenders higher and more prolonged immunity [32,33]. Future research that unravels and compares the immunological mechanism of between homologous and heterologous prime–booster regimens is of huge academic value.

Regarding the methodology, this study contains both strengths and limitations. The use of routine nationwide service data is one of the key strengths since the findings can truly reflect the real-world vaccine effectiveness in the backdrop of the day-to-day health system performance. Yet, some limitations remain. First, the study relied on secondary data from different sources, each of which had its own data collecting protocol. During the data merging process, some information, which was not collected across the board, was dropped, such as occupation profiles, underlying diseases, and risk history of each individual. Therefore, residual confounding may still exist. To address this concern, the matching by age, living province, and time of specimen collection helped minimize these confounding effects. Moreover, the findings from restricted samples (high-risk participants) were quite similar to the main analysis. This implies that individual infection risk was soundly controlled by the matching process. Second, information bias cannot be evaded since the identification of cases was performed by either PCR or Ag-RDT. Ag-RDT is widely acknowledged for the inferior test performance, particularly sensitivity, relative to PCR. Hence, misclassification of infection status might occur. As the non-vaccinee group might have a larger fraction of severe cases, and owing to the admission protocol for many Thai hospitals, the severe cases were obliged to undertake PCR prior to admission, it is possible that our VE estimate was underestimated. Moreover, the VE estimate might be diluted if the proportion of participants undergoing Ag-RDT did not vary much by vaccination status, especially among asymptomatic or mild cases (non-differential misclassification bias). However, the volume of cases identified solely by Ag-RDT in the Co-Lab system was still far lower than PCR confirmed cases, and we included only professional-use Ag-RDT while excluding self-test Ag-RDT. Therefore, such bias might not severely compromise the result validity and the potential marginal underestimation of the effect suggested that the true VE might be even higher than the values observed in this study. Last, the measures gained from test-negative case control design do not always reflect those acquired from population-based case control design. It is universally accepted that the true number of COVID-19 cases is under-reported as some infectees are asymptomatic or have very mild symptoms, making them unaware of their infective status. In other words, the cases identified by the Co-Lab system do not necessarily mirror the true case volume in the population. Nonetheless, we deemed that the test-negative design is practically valid for studying VE in this context since the design has key advantages in controlling for similar participation rates, initial presentation, and diagnostic suspicion tendencies between cases and controls [34].

## 5. Conclusions

Though the degree of protection against any infection varied across vaccine regimens, all regimens revealed favorable effects against severe infection. As the effectiveness of two-dose regimens declined over time, a third-dose booster shot plays critical role for a country to achieve population herd immunity. The mix-and-match of vaccine regimens demonstrated acceptable outcomes with regard to the protection against both any and severe infections. Viral vector vaccine followed by mRNA vaccine exhibited the greatest protection level. Heterologous prime-boost regimens should be considered as an alternative to address vaccine shortage and accelerate the national vaccine rollout plan. Further monitoring on the effectiveness of various vaccine regimens while accounting for the advent of many more SARS-CoV-2 variants in the future is recommended.

## Figures and Tables

**Figure 1 vaccines-10-01080-f001:**
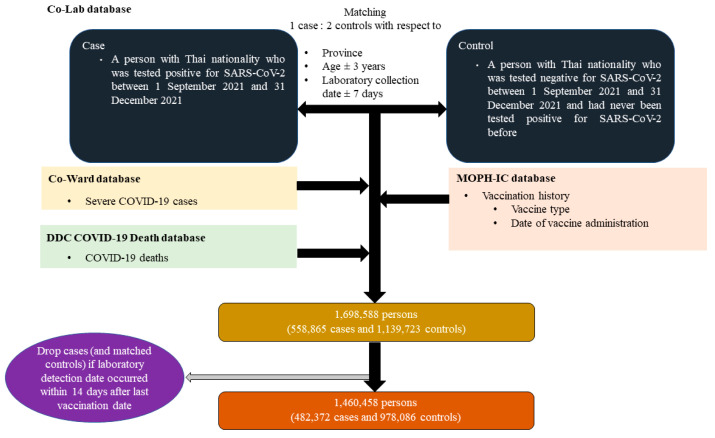
Flow diagram of data retrieving process.

**Figure 2 vaccines-10-01080-f002:**
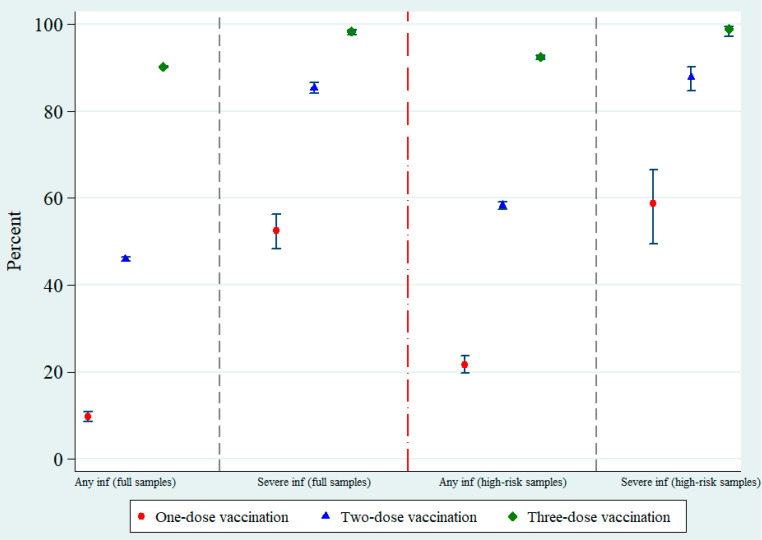
Vaccine effectiveness against any infection and severe infection for each vaccination status (any combination of vaccine brands). Note: Vertical bars denote 95% confidence interval; *p* < 0.001 for all schedules.

**Figure 3 vaccines-10-01080-f003:**
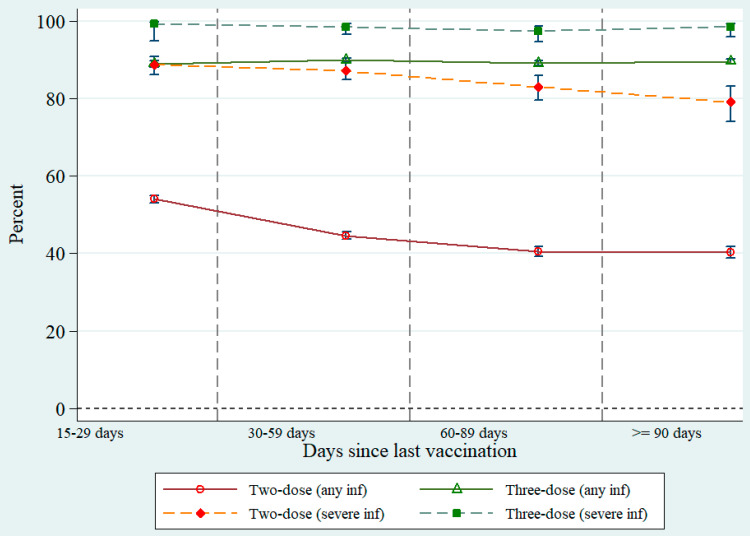
Vaccine effectiveness against any infection and severe infection among two- and three-dose vaccinees over time (any combination of vaccine brands). Note: Vertical bars denote 95% confidence interval; *p* < 0.001 for all schedules.

**Table 1 vaccines-10-01080-t001:** Characteristics of the overall study participants.

Characteristics	All—*n* (%) (*N* = 1,698,588)	Cases Sorted by Severity—*n* (%) (*N* = 558,865)		Controls—*n* (%) (*N* = 1,139,723)
		Non-Severe (*N* =548,745)	Severe (*N* = 10,120)	
Age groups—years				
<18	135,392 (8.0)	47,030 (8.6)	64 (0.6)	88,298 (7.7)
18–59	1,352,447 (79.6)	442,627 (80.6)	4982 (49.2)	904,838 (79.4)
>60	210,749 (12.4)	59,088 (10.8)	5074 (50.2)	146,587 (12.9)
High-risk samples				
No	967,267 (57.0)	327,102 (59.6)	6659 (65.8)	633,506 (55.6)
Yes	731,231 (43.0)	221,643 (40.4)	3461 (34.2)	506,217 (44.4)
Regions of testing				
Upcountry	1,376,258 (81.0)	450,401 (82.1)	8085 (79.9)	917,772 (80.5)
Greater Bangkok	322,330 (19.0)	98,344 (17.9)	2035 (20.1)	221,951 (19.5)

**Table 2 vaccines-10-01080-t002:** Vaccination status of cases and controls included in the conditional logistic regression.

Characteristics	All—n (%) (*N* = 1,460,458)	Cases Sorted by Severity—*n* (%) (N = 482,372)		Controls—*n* (%) (*N* = 978,086)
		Non-Severe (*N* =473,433)	Severe (*N* = 8939)	
No vaccination	529,128 (36.2)	205,174 (43.3)	5903 (66.1)	318,051 (32.5)
One-dose vaccination	240,781 (16.5)	88,487 (18.7)	1635 (18.3)	150,659 (15.4)
Two-dose vaccination	562,667 (38.5)	169,239 (35.8)	1353 (15.1)	392,075 (40.1)
Three-dose vaccination	127,882 (8.8)	10,533 (2.2)	48 (0.5)	117,301 (12.0)

**Table 3 vaccines-10-01080-t003:** Vaccine effectiveness against any infection for two-dose and three-dose vaccine regimens sorted by time since last vaccination.

Regimen	15–29 Days	30–59 Days	60–89 Days	90 Days Onward
Two-dose				
BNT162b2 + BNT162b2	74.2 ** (71.8–76.3)	62.4 ** (59.4–65.1)	69.1 ** (64.3–73.2)	57.0 ** (43.6–67.2)
ChAdOx1 + ChAdOx1	61.4 ** (59.6–63.2)	50.3 ** (48.3–52.3)	43.2 ** (40.0–46.2)	25.8 ** (19.1–31.9)
CoronaVac + CoronaVac	27.9 * (0.3–47.9)	52.2 ** (43.6–59.5)	55.2 ** (52.4–57.9)	49.8 ** (47.8–51.6)
ChAdOx1 + BNT162b2	79.9 ** (74.0–84.5)	88.7 ** (85.6–91.1)	89.0 ** (84.7–92.0)	77.4 ** (68.2–84.0)
CoronaVac + ChAdOx1	57.8 ** (56.3–59.2)	47.5 ** (46.2–48.8)	40.5 ** (38.5–42.2)	36.6 ** (33.6–39.4)
CoronaVac + BNT162b2	74.7 ** (62.8–82.8)	82.3 ** (74.3–87.7)	85.7 ** (74.6–92.0)	84.6 ** (64.9–89.3)
Three-dose				
CoronaVac + CoronaVac + ChAdOx1	87.1 ** (85.2–88.8)	87.4 ** (86.2–88.4)	87.5 ** (86.2–88.6)	89.2 ** (88.0–90.3)
CoronaVac + CoronaVac + BNT162b2	95.0 ** (93.8–95.9)	94.0 ** (93.4–94.6)	91.8 ** (90.9–92.6)	90.6 ** (89.4–91.6)

Note: Figures in parenthesis denote 95% confidence interval; * *p* < 0.05; ** *p* < 0.001.

**Table 4 vaccines-10-01080-t004:** Vaccine effectiveness against severe infection for two-dose and three-dose vaccine regimens sorted by time since last vaccination.

Regimen	15–29 Days	30–59 Days	60–89 Days	90 Days Onward
Two-dose				
BNT162b2 + BNT162b2	86.4 *** (65.3–94.7)	86.7 *** (68.8–94.3)	91.3 ** (63.0–97.9)	NA
ChAdOx1 + ChAdOx1	88.2 * (83.1–91.8)	87.8 ** (83.3–91.1)	81.5 *** (73.9–87.9)	80.9 *** (67.8–88.7)
CoronaVac + CoronaVac	NA	85.9 ** (38.7–96.8)	88.9 *** (77.9–94.4)	80.3 *** (72.0–86.2)
ChAdOx1 + BNT162b2	92.1 * (39.2–99.0)	92.0 ** (66.3–98.1)	NA	NA
CoronaVac + ChAdOx1	92.8 *** (89.5–95.2)	91.7 *** (88.8–93.8)	87.8 *** (83.0–91.3)	84.4 *** (75.7–88.7)
CoronaVac + BNT162b2	NA	NA	NA	NA
Three-dose				
CoronaVac + CoronaVac + ChAdOx1	NA	99.1 *** (93.6–99.9)	97.9 *** (91.4–99.5)	NA
CoronaVac + CoronaVac + BNT162b2	97.2 *** (79.8–99.6)	97.6 *** (90.3–99.4)	98.9 *** (92.1–99.8)	98.7 *** (90.6–99.8)

Note: Figures in parenthesis denote 95% confidence interval; * *p* < 0.05; ** *p* < 0.01; *** *p* < 0.001; NA refers to “not applicable”.

## Data Availability

Not applicable.

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
