# Peer review of "Real-World Effectiveness of Mix-and-Match Vaccine Regimens against SARS-CoV-2 Delta Variant in Thailand: A Nationwide Test-Negative Matched Case-Control Study"

_vaccines, 2022, doi:10.3390/vaccines10071080_

Round 1

Reviewer 1 Report

This work includes a dataset derived from a nationwide registry in Thailand. Thus, the authors confirm that heterologous vaccine regimens lead to better outcomes compared to homologous vaccines. In addition, a third dose shows the best performance.

The data are well presented.  It is further advice on how SARS-CoV-2 vaccination should be managed by an infectious disease specialist.

The discussion is somewhat lengthy, but all data are well embedded in the current literature. 

Author Response

This work includes a dataset derived from a nationwide registry in Thailand. Thus, the authors confirm that heterologous vaccine regimens lead to better outcomes compared to homologous vaccines. In addition, a third dose shows the best performance. The data are well presented.  It is further advice on how SARS-CoV-2 vaccination should be managed by an infectious disease specialist. The discussion is somewhat lengthy, but all data are well embedded in the current literature.

  • Thank you

Reviewer 2 Report

Interesting manuscript with minor corrections needed. Table 2 percentage not tallying to 100%.

Author Response

Interesting manuscript with minor corrections needed. Table 2 percentage not tallying to 100%.

  • Thank you. The term 5.8% is now changed to 8.8%.
  • All grammatical errors are corrected.

Reviewer 3 Report

This paper presents an interesting epidemiological study presenting data about the effectiveness of various mix and match vaccine regimens against any and severe COVID infection, in the population of Thailand. The authors (and the national administration) are to be commended for their capacity to dispose of such complete and exhaustive information for the whole country, organised in systematic databases with cross-identification of individuals possible based on the national identity. This is a powerful epidemiological tool, and of particular value in times of a crisis such as the present COVID pandemic.

The data are of use to public health deciders, as the authors describe good vaccine effectiveness (VE) with mixed vaccines, and very high VE with three-dose schedules. These data provide a sound rationale for vaccinating populations with whatever vaccine is available, without worrying about using the same vaccine as given at a previous dose.

Overall, I have no major remarks. Despite the limitations, which are well explained by the authors, the sheer volume of individuals involved in the analysis confers good reliability on the data, and the analyses have been performed appropriately.

There are many minor grammar mistakes throughout the paper; I would advise having it read by a native English speaking medical writer. Below are some minor suggestions for correction.

-          Page 2 line 46 (and page 9, line 260) – you cannot use the term “topple down” in this sense. Please replace by “reduce”.

-          Page 2, line 57, please change “evidence suggested” to “evidence suggesting”

-          Page 4, line 136, please replace “a dead case” by “death”

-          Page 4, line 161, please change “left in the pool” to “remained in the analysis”

-          Page 5, line 172, please change “from the same proportion in September” to simply “from the proportion in September” (It is not the same, since it was 13.1% in December, versus 6.2% in September).

-          Page 9, line 247, please change “Albeit” to “Although”

-          Page 9, line 248, please change “mundane” to “mediocre”

-          Page 9, line 261, please change “individual discrepancy” to “individual propensity”

-          Page 9, lines 289/290, please change “despite not so many in number” to “although few in number”

-          Page 10, line 311, please change “unpacks” to “unravels”

-          Page 10, line 313, “methodology-wise” is too colloquial. Please change to “Regarding the methodology, …”

-          Page 10, line 319, remove the word “out” (it should read “dropped”)

-          Page 10, line 320, remove the s on the word “confoundings” (confounding is singular)

-          Page 10, lines 330/331 – I am unable to understand what the authors mean by the sentence in parenthesis from “(let alone…..” down to “….mild cases).” This should be rephrased.

-          Page 10, line 336, please change the word “implied” to “given”

-          Page 10, line 340, please change “pertain” to “have”

-          Page 10, line 341, please change “unrealize” to “unaware”

-          Page 10, line 342, please change to “….Co-Lab system do not necessarily…” (the subject of this sentence is “the cases”, therefore plural)

-          Page 10, line 345, remove the word “of”

-          Page 11, line 354, please put “Heterologous prime-boost regimens” plural.

Author Response

This paper presents an interesting epidemiological study presenting data about the effectiveness of various mix and match vaccine regimens against any and severe COVID infection, in the population of Thailand. The authors (and the national administration) are to be commended for their capacity to dispose of such complete and exhaustive information for the whole country, organised in systematic databases with cross-identification of individuals possible based on the national identity. This is a powerful epidemiological tool, and of particular value in times of a crisis such as the present COVID pandemic. The data are of use to public health deciders, as the authors describe good vaccine effectiveness (VE) with mixed vaccines, and very high VE with three-dose schedules. These data provide a sound rationale for vaccinating populations with whatever vaccine is available, without worrying about using the same vaccine as given at a previous dose.

  • Thank you

Overall, I have no major remarks. Despite the limitations, which are well explained by the authors, the sheer volume of individuals involved in the analysis confers good reliability on the data, and the analyses have been performed appropriately. There are many minor grammar mistakes throughout the paper; I would advise having it read by a native English speaking medical writer. Below are some minor suggestions for correction.

  • Thank you, we have re-checked all the spellings again before resubmitting the manuscript.

-          Page 2 line 46 (and page 9, line 260) – you cannot use the term “topple down” in this sense. Please replace by “reduce”.

  • Revised, line 47.

-          Page 2, line 57, please change “evidence suggested” to “evidence suggesting”

  • Revised, line 58.

-          Page 4, line 136, please replace “a dead case” by “death”

  • Revised, line 138.

-          Page 4, line 161, please change “left in the pool” to “remained in the analysis”

  • Revised, line 164.

-          Page 5, line 172, please change “from the same proportion in September” to simply “from the proportion in September” (It is not the same, since it was 13.1% in December, versus 6.2% in September).

  • Revised, line 175.

-          Page 9, line 247, please change “Albeit” to “Although”

  • Revised, line 250.

-          Page 9, line 248, please change “mundane” to “mediocre”

  • Revised, line 251.

-          Page 9, line 261, please change “individual discrepancy” to “individual propensity”

  • Revised, line 265.

-          Page 9, lines 289/290, please change “despite not so many in number” to “although few in number”

  • Revised, line 297.

-          Page 10, line 311, please change “unpacks” to “unravels”

  • Revised, line 319.

-          Page 10, line 313, “methodology-wise” is too colloquial. Please change to “Regarding the methodology, …”

  • Revised, line 321.

-          Page 10, line 319, remove the word “out” (it should read “dropped”)

  • Revised, line 327.

-          Page 10, line 320, remove the s on the word “confoundings” (confounding is singular)

  • Revised, line 328.

-          Page 10, lines 330/331 – I am unable to understand what the authors mean by the sentence in parenthesis from “(let alone…..” down to “….mild cases).” This should be rephrased.

  • Revised, line 336-339.

-          Page 10, line 336, please change the word “implied” to “given”

  • We changed to the term “suggested”, line 345.

-          Page 10, line 340, please change “pertain” to “have”

  • Revised, line 349.

-          Page 10, line 341, please change “unrealize” to “unaware”

  • Revised, line 350.

-          Page 10, line 342, please change to “….Co-Lab system do not necessarily…” (the subject of this sentence is “the cases”, therefore plural)

  • Revised, line 351.

-          Page 10, line 345, remove the word “of”

  • Revised, line 354.

-          Page 11, line 354, please put “Heterologous prime-boost regimens” plural.

  • Revised, line 363.